# SymMatika: Structure-Aware Symbolic Discovery

## Abstract

Symbolic regression (SR) seeks to recover closed-form mathematical expressions that describe observed data. While existing methods have advanced the discovery of either explicit mappings (i.e., $y = f(\mathbf{x})$) or discovering implicit relations (i.e., $F(\mathbf{x}, y) = 0$), few modern and accessible frameworks support both. Moreover, most approaches treat each expression candidate in isolation, without reusing recurring structural patterns that could accelerate search. We introduce SymMatika, a hybrid SR algorithm that combines multi-island genetic programming (GP) with a reusable motif library inspired by biological sequence analysis. SymMatika identifies high-impact substructures in top-performing candidates and reintroduces them to guide future generations. Additionally, it incorporates a feedback-driven evolutionary engine and supports both explicit and implicit relation discovery using implicit-derivative metrics. Across benchmarks, SymMatika achieves state-of-the-art recovery rates on the Nguyen and Feynman benchmark suites, an impressive recovery rate of 61% on Nguyen-12 compared to the next best 2%, and strong placement on the error-complexity Pareto fronts on the Feynman equations and on a subset of 57 SRBench Black-box problems. Our results demonstrate the power of structure-aware evolutionary search for scientific discovery. To support broader research in interpretable modeling and symbolic discovery, we have open-sourced the full SymMatika framework.

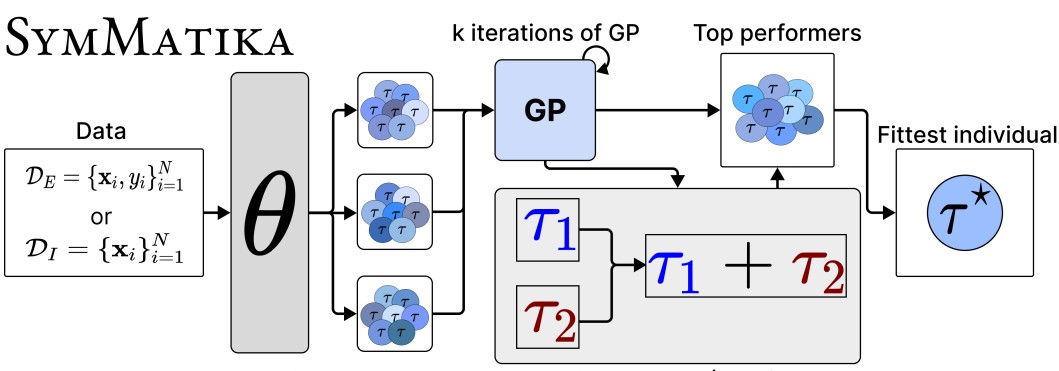

Fig. 1: **SymMatika** is a high-performing symbolic regression framework. Given data $\mathcal{D}_E$ (explicit) or $\mathcal{D}_I$ (implicit), a parameterized generator $\theta$ produces $m$ initial populations. Each population evolves for $k$ GP iterations, and the top $M$ expressions are used to update both the generator parameters $\theta$ and a reusable structure (i.e., Motif) library. High-impact subexpressions are recombined to form new candidates to accelerate convergence. The fittest individual $\tau^\star$ is selected as the final expression.

## 1 Introduction

In the late 16th century, Tycho Brahe meticulously recorded the positions of celestial bodies. His data laid the foundation for Johannes Kepler, who in 1601 derived analytical expressions of the motion of these planets. These expressions launched a scientific revolution in their discovery that Mars'

orbit was in fact an ellipse. This was an early instance of *symbolic regression* (SR), where governing laws are distilled from observational data as interpretable mathematical expressions. The majority of SR methods similarly aim to discover symbolic expressions $f$ that relate variables in data $(\mathbf{X}, y)$, with $X_i \in \mathbb{R}^n$, $y_i \in \mathbb{R}$, in the form of $y = f(\mathbf{x})$. Some others seek to uncover implicit relations $F(\mathbf{x}, y) = 0$ to discover invariants such as conservation laws.

Symbolic regression poses a computationally hard challenge, as the space of symbolic expressions grows exponentially with expression length (Lu et al., 2016; Virgolin & Pissis, 2022). The dominant approach, genetic programming (GP) (Koza, 1990), stochastically evolves populations of candidate expressions using mutation and crossover. Tools such as PySR (Cranmer, 2023), GPlearn (Stephens et al., 2016), Operon (Burlacu et al., 2020), and AFP (Schmidt & Lipson, 2010) apply GP to flexibly search across diverse mathematical structures without assuming parametric forms. However, their convergence is often slow due to uninformed exploration, generating many unpromising candidates across generations.

To improve efficiency, neural-guided SR approaches integrate deep learning to bias the search toward promising expressions. Methods such as DSR (Petersen et al., 2019), AI Feynman (Mundhenk et al., 2021), and NGGPPS (Udrescu & Tegmark, 2020) use RNNs or Transformers to guide expression generation or predict symbolic transformations. Deep learning and GP-based approaches are alternative methods of symbolic regression. We make comparisons between GP-based and deep-learning algorithms on benchmark performances later in Sec. 4.

Recent algorithms (Petersen et al., 2019; Mundhenk et al., 2021) start to use recurrent neural networks (RNNs) to learn trends in expression coefficients. However, they do not capture higher-order syntactic structures, such as the recurring mathematical motifs found in top-performing individuals. Therefore, these approaches miss the opportunities to build complex expressions from learned substructures. Identifying and reusing these structures could accelerate discovery by enabling recombination of partial solutions into globally correct forms. Moreover, most existing SR systems specialize in either explicit mappings or implicit relations, but not both. The only known system supporting both, Eureqa (Schmidt & Lipson, 2009), has remained closed sourced and now is integrated into a commercialized platform.

In this work, we introduce SYMMATIKA, a unified symbolic regression framework that discovers both explicit and implicit mathematical relations by combining multi-population feedback-based genetic programming with learned structural pattern reuse. Given data $\mathcal{D}_E = \{(\mathbf{x}_i, y_i)\}_{i=1}^N$ for explicit tasks or $\mathcal{D}_I = \{(\mathbf{x}_i)\}_{i=1}^N$ for implicit ones, SYMMATIKA iterates over three key phases: (1) feedback-based GP, which adaptively adjusts operation weights based on evolutionary context; (2) structural analysis, which extracts frequent syntactic substructures from high-performing expressions; and (3) population-specific updates to generation parameters, enabling structurally guided tree construction. The algorithm returns the fittest expression across all populations. In experiments on the Nguyen benchmark and Feynman equations, SYMMATIKA outperforms state-of-the-art methods in accuracy and convergence. It also recovers implicit governing equations from experimental data in the Eureqa dataset with up to $100\times$ faster, demonstrating both versatility and performance.

## 2 RELATED WORK

**Genetic Programming for Symbolic Regression.** GP has long been a cornerstone of symbolic regression, originating with Koza's foundational work (Koza, 1990; 1994), which introduced the idea of evolving expression trees via biologically inspired operations such as mutation and crossover. Modern GP-based SR systems such as Operon (Burlacu et al., 2020) and PySR (Cranmer, 2023) inherit this lineage. Operon uses a steady-state GP model with tournament selection, while PySR employs a multi-population island model and simulated annealing to balance exploration and exploitation during evolution. These frameworks rely on fixed operator probabilities, which can hinder adaptive search in complex landscapes. Feedback-based adaptive crossover-rate in evolutionary computation (Guan et al., 2024) proposes a modifiable crossover distribution to optimize crossover points, however it does not control the probability of crossover being selected over other genetic operations (i.e. mutation or single-node crossover). Self-adjusting mutation rates with provably optimal success rules (Doerr et al., 2019) proposes updating mutation rates based on fitness-success of offspring, yet it does not consider the overall evolutionary progress of the algorithm.

SYMMATIKA builds on this GP foundation but introduces two key innovations. First, it uses *feedback-based operator scheduling* that dynamically adjusts mutation, crossover, and selection rates based on each population's recent evolutionary progress (Sec. 3.2). Second, it augments the search with a reusable *structural motif library*, extracted from high-performing expressions across populations and generations (Sec. 3.3). This enables the recombination of semantically meaningful substructures, promoting convergence toward globally correct solutions. Unlike previous GP models that operate solely at the token or subtree level with fixed heuristics, SYMMATIKA evolves both structural and operator-level strategies in tandem.

**Neural-Guided Symbolic Regression.** Neural-guided SR emerged to improve search efficiency by integrating neural networks into symbolic discovery (Martius & Lampert, 2016; Alaa & Van der Schaar, 2019; Kamienny et al., 2022; Biggio et al., 2021; Champion et al., 2019). Among these, AI Feynman (Udrescu & Tegmark, 2020) is notable for using neural networks to detect symmetries, units, and separability in the data, enabling recursive decomposition of complex equations. More directly comparable to our method is DSO Mundhenk et al. (2021), which combines an RNN-based generator with stateless, random-restart GP loops. Their model generates $N$ candidate expressions per iteration, refines them over $S$ GP steps, and uses the best $M$ expressions to update the RNN via policy gradients.

While both our SYMMATIKA and DSO use learning-based components to guide symbolic search, our approach diverges in two critical ways. First, instead of training a monolithic RNN to guide sampling, we perform population-specific frequency analysis over high-performing expressions to update generation parameters, enabling interpretable and efficient adaptation. Second, and more fundamentally, DSO learns only at the *token level* (e.g., which operators and constants are promising), whereas SYMMATIKA extracts and reuses high-impact *structural patterns* – subtrees that recur in successful individuals. This is especially useful to recombine partial symbolic solutions into novel candidates for full solutions, facilitating both exploration and repair. Furthermore, our focus is orthogonal to recent work using deep neural networks to uncover latent variables (Chen et al., 2022), equations (Brunton et al., 2016), or structures (Huang et al., 2024). These approaches aim to extract informative features from data. In fact, these approaches can integrate symbolic regression into their process of discovering governing principles.

**Eureqa and Implicit Symbolic Regression.** Most modern SR systems focus on discovering explicit functional mappings $y = f(\mathbf{x})$, while implicit relations $F(\mathbf{x}, y) = 0$, commonly seen in physical systems governed by conservation laws or symmetries, remain underexplored. PIE (Yufei et al., 2025) treats implicit regression as a translation problem by mapping point-set data to symbolic skeletons, using priors learned through supervised pre-training on explicit regression equations. However, PIE relies on supervised data for pre-training, while SYMMATIKA requires no priors or supervised pre-training, and can immediately symbolically regress on unsupervised data.

Eureqa (Schmidt & Lipson, 2009) is another of the few frameworks capable of discovering both explicit and implicit expressions. It uses a Pareto-optimized GP framework that perturbs, recombines, and simplifies expressions, scoring candidates via error and complexity. Importantly, Eureqa incorporates implicit-derivative metrics to recover nontrivial invariant equations.

Despite its versatility, Eureqa is over 16 years old and struggles on modern benchmarks due to fixed operator schedules and heuristics tailored for low-dimensional problems (originally $\leq 4$ variables). SYMMATIKA preserves Eureqa's implicit-derivative loss but augments it with improved GP techniques including feedback-driven operator tuning, structural motif reuse, and multi-population coordination. These enhancements enable SYMMATIKA to recover implicit equations up to $100\times$ faster and achieves superior performance on explicit SR benchmarks such as Feynman and Nguyen.

## 3 SYMMATIKA

SYMMATIKA is composed of two core components: (1) a multi-population, feedback-driven genetic programming engine, and (2) a library of high-impact symbolic motifs that capture reusable substructures. In this section, we introduce the formal setup and describe each component, followed by their integration for discovering both explicit and implicit expressions in data.

## 3.1 Problem Setup

We represent symbolic expressions as algebraic trees $\tau$, with internal nodes as unary or binary operators (e.g., $+$, $\times$, $\cos$, $\log$) and leaf nodes as constants or variables. A pre-order traversal of $\tau$ yields the symbolic expression $f$, whose quality is assessed using task-specific fitness metrics.

**Explicit relations.** Given data $\mathcal{D}_E = \{(\mathbf{x}_i, y_i)\}_{i=1}^N$ with $\mathbf{x}_i \in \mathbb{R}^d$, the goal is to find $f$ such that $y \approx f(\mathbf{x})$. We define fitness as mean-log-error (MLE):

$$\mathcal{L}_{\mathcal{D}_E}(f) = -\frac{1}{N}\sum_{i=1}^N \log\left(1 + |y_i - f(\mathbf{x}_i)|\right)$$

**Implicit relations.** When target variables are not explicitly labeled, as is common in physical systems governed by invariants, we aim to discover expressions $f(\mathbf{x}) = 0$ that characterize underlying constraints or symmetries. Given a candidate expression $f(x_1, x_2, \ldots, x_n) = 0$, to quantify fitness, we apply the implicit function theorem. Treating $x_i$ as an implicit function of $x_j$, the derivative is given by:

$$\frac{\partial x_i}{\partial x_j} = -\frac{\frac{\partial f}{\partial x_j}}{\frac{\partial f}{\partial x_i}}$$

We compute symbolic partial derivatives across all variable pairs and compare them against finite-difference numerical estimates. To account for variable interdependencies, which frequently occur in coupled dynamical systems, we generalize the paired partial derivative as:

$$\frac{\partial x_i}{\partial x_j} = \frac{\partial x_i + \partial x_p \cdot \frac{\Delta x_p}{\Delta x_i}}{\partial x_j + \partial x_q \cdot \frac{\Delta x_q}{\Delta x_j}}$$

where $x_p$ and $x_q$ are variables interdependent with $x_i$ and $x_j$, respectively. $\Delta$ represents finite differences in numerical analysis (including time-series data), or the numerical approximation of the derivative, and $\partial$ represents the calculus partial derivative of a symbolic expression. We evaluate all possible such pairings and take the worst-case pairing for evaluation to penalize expressions that perform well only under selective dependencies.

Let $M_s$ and $M_n$ denote the symbolic and numerical paired-partial derivative matrices (shape $\binom{d}{2} \times N$). The implicit fitness is:

$$\mathcal{L}_{\mathcal{D}_I}(f) = -\frac{1}{N}\sum_{i=1}^N \log\left(1 + \|M_n(\mathbf{x}_i) - M_s(\mathbf{x}_i)\|\right)$$

## 3.2 Feedback-Based Genetic programming

Tree-based genetic programming uses two core operators: *crossover*, which swaps subtrees at selected nodes between parents, and *mutation*, which perturbs nodes within an individual. These are typically coupled with a selection mechanism (e.g., tournament selection) that favors high-fitness candidates. While effective, traditional GP applies static operator rates and rigid selection rules, often overlooking valuable substructures in lower-fitness individuals and failing to adapt to population dynamics. To address these limitations, we extend GP with four key mechanisms:

**Single-node crossover.** Traditional subtree crossover introduces large structural changes. To enable finer control during late-stage optimization, we introduce single-node crossover, which swaps only one same-type node between trees (i.e. binary operation $\leftrightarrow$ binary operation, unary operation $\leftrightarrow$ unary operation, variable $\leftrightarrow$ variable, etc.).

**Temperature-guided selection and mutation.** Simulated annealing is a well-established strategy for balancing exploration and exploitation in evolutionary algorithms. PySR (Cranmer, 2023) incorporates this by rejecting a mutation with probability $p = \exp\left(\frac{L_F - L_E}{\alpha T}\right)$, where $L_E$ and $L_F$ are the fitness scores of an expression before and after mutation, $T \in [0, 1]$ is the annealing temperature, and

$\alpha$ is a scaling hyperparameter. While effective for mutation acceptance, PySR continues to rely on tournament selection, which prioritizes only the fittest individuals, favoring exploitation over exploration.

To promote further exploration of potentially promising but lower-fitness individuals, we extend simulated annealing to the selection mechanism itself. During selection, we randomly sample a subset of individuals from a population and assign each a Boltzmann probability: $p_f = \exp(\frac{\mathcal{L}_{\mathcal{D}}(f)}{T})$ where $\mathcal{L}_{\mathcal{D}}(f)$ is the fitness of candidate $f$. Selection is then performed via roulette sampling over normalized probabilities. At high temperatures ($T \to 1$), selection approximates uniform sampling, promoting exploration. As $T$ decreases, the probability mass concentrates on high-fitness candidates, effectively converging toward tournament-style selection. This adaptive strategy allows the selection mechanism to gradually shift from exploration to exploitation as the evolutionary process progresses.

We apply a similar temperature-dependent strategy to mutation. At high temperatures, coarse mutations such as subtree replacement are favored to encourage diversity. As $T$ lowers, finer mutations, such as constant perturbations or operator swaps, are more likely. Each population $P_i$ maintains a temperature-adjusted distribution over mutation types, enabling population-specific tuning of structural granularity. Together, these temperature-guided mechanisms provide principled control over both the scope of variation and the selective pressure during evolution.

**Feedback-based operator scheduling.** Traditional GP systems use fixed rates for genetic operators such as mutation and crossover (Schmidt & Lipson, 2009). While simple, static rates are suboptimal: coarse-grained changes like subtree crossover are useful early in evolution, whereas fine-grained adjustments like coefficient tuning are better suited for later stages. To enable adaptive behavior, we introduce feedback-based scheduling that dynamically adjusts operator probabilities based on the evolutionary context of each population.

Let $g_0$ be the average fitness of the top-$M$ individuals in a population prior to GP loop, and let $g_n$ be the same statistic after $n$ generations. Let $h$ denote the number of consecutive generations with negligible fitness improvement (plateau), defined as $|g_n - g_{n-1}| < \epsilon$ for some threshold $\epsilon = 1e^{-6}$. We define the operator probability function:

$$\mathbb{P}(g_n, h) = \begin{cases} m_i \pm (\frac{|g_0 - g_n|}{g_0} \cdot |m_f - m_i|) \pm 0.1h & h \geq 2 \\ m_i \pm (\frac{|g_0 - g_n|}{g_0} \cdot |m_f - m_i|) & \text{else} \end{cases}$$

Here, $m_i$ and $m_f$ are the initial and final operator probabilities for each genetic operation (e.g. crossover, mutation), with signs determined by whether each operator frequency should increase or decrease over time. Crossover starts with probability 60% and decreases to 5%, single-node crossover starts with probability 10% and increases to 15%, mutation starts with probability 30% and increases to 80%. This function is safe-guarded by these probability bounds and our probabilistic selection model, so it is resilient to abnormal degradations in average fitness.

**Island populations with migration.** We evolve a set of populations or "islands" in parallel. The island model for GP is a powerful tool for simultaneously evolving multiple distinct evolutionary paths (Duarte et al., 2017; Whitley et al., 1999). We allow for small migrations between islands by swapping small subsets of individuals between populations. Initially, 1% of island populations are swapped every 20 iterations. These subsets increase to 2% of island populations according to a growing control rate $\alpha_M$. We keep migrations small and infrequent to promote population diversity and prevent convergent evolution, which we tend to observe when migration rates exceed 3%.

### 3.3 Motif Library and Structure Reuse

After each iteration of GP, we generate new candidates to replace subsets in each population $P$ of worst-performing individuals. Candidate generation in SYMMATIKA is modeled as a parameterized distribution over expression trees, denoted $p(\tau \mid \theta)$, where $\tau$ is a symbolic expression and $\theta$ encodes node-level generation probabilities (e.g., for selecting constants, variables, and operators) (Mundhenk et al., 2021; Petersen et al., 2019). These parameters are updated based on frequency feedback from high-fitness individuals: for each operator type $\theta_i$, we compute its frequency $f_{\theta_i}$ across top-performing expressions, and apply the update:

$$\theta_i \leftarrow \theta_i + \beta_{\theta_i} f_{\theta_i}$$

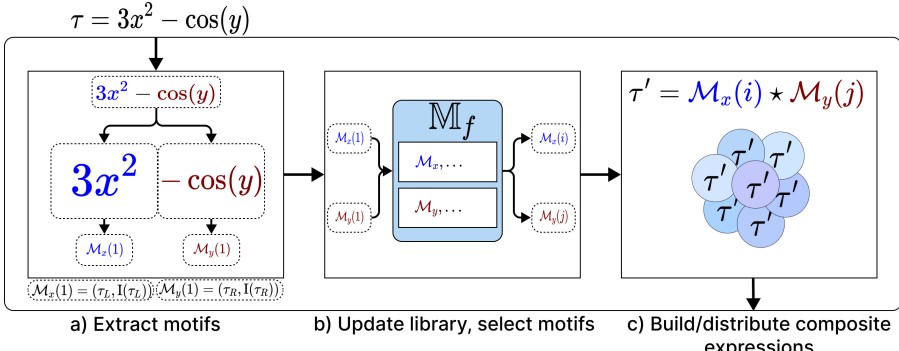

Fig. 2: Motif-based recombination across evolving populations with shared top performers.

Here, $\beta_{\theta_i}$ is a scaling factor that accounts for natural biases, e.g., binary operators ($\beta_{\theta_i} = 0.2$) such as $+$ and $\times$ may occur more often than unary functions ($\beta_{\theta_i} = 1.0$) like $\log$ or $\exp$, and are thus adjusted more conservatively. These biases were selected from observations and tuning of our parameterized candidate generator during initial testing. This feedback mechanism enables population-specific adaptation of the generative distribution, which can bias expression construction toward promising token-level patterns.

However, frequency-based feedback alone is limited: it captures which tokens are useful, but cannot be used to explicitly discover structural patterns or partial solutions in individuals. For instance, in a symbolic regression problem with correct solution $f = x^2 + y^2 + z^2$, a partial solution $f' = x^2 + y^2 - \cos(z)$ would not be recognized as having some correct terms through frequency analysis. To better identify and reuse partial solutions, we introduce a structural motif discovery mechanism inspired by biological sequence motifs.

In molecular biology, a sequence motif is a recurring nucleotide or amino-acid pattern with functional significance (Tateno et al., 1997; Liu et al., 2002; Chou & Schwartz, 2011; Grant & Bailey, 2021). Analogously, we define a symbolic motif as a high-impact subexpression, a subtree within an expression that contributes strongly to its overall fitness. We use this intuition to construct a reusable library of symbolic motifs and employ them for guided recombination.

**Initialization and Data Structure.** To implement structural reuse, we maintain a central motif library $\mathbb{M}_f$ that stores high-impact symbolic subexpressions across populations. This library is structured as a $d \times k$ table, where $d$ corresponds to the number of data variables and each row holds up to $k$ motifs associated with a particular variable, where $k$ is typically a small integer ($\leq 20$). Each entry in the table consists of a motif $\mathcal{M}_v = (\tau', I(\tau'))$, where $\tau'$ is a symbolic subtree and $I(\tau')$ is its estimated impact score on fitness. We calculate impact score with the following equation $I(\tau')) = \mathcal{L}(\tau) - \mathcal{L}(\tau - \tau')$, where $\mathcal{L}(\tau)$ is the loss of the original tree expression and $\mathcal{L}(\tau - \tau')$ is the loss of the original tree without the subtree $\tau'$. This quantifies the importance of the substructure to the overall quality of the expression.

**Motif Generation.** After each generation, we examine the top $M$ individuals from each island population to identify candidate motifs. For a given expression $\tau$ (e.g., $\tau = 3x^2 - \cos(y)$), we extract the left and right subtrees of each internal node (e.g., $\tau_L = 3x^2$, $\tau_R = -\cos(y)$. Note: $\cos(y)$ is a negative term in the example, so we extract it as $-\cos(y)$) and assess their fitness contribution.

Each motif is then associated with a specific input variable $v$ by scanning the subtree in pre-order and assigning it to the first variable encountered. This heuristic is based on the assumption that symbolic components are typically rooted in a dominant variable, and indexing motifs by variable helps maintain coverage and diversity during recombination. The newly extracted motifs are compared to the existing entries in row $v$ of $\mathbb{M}_f$. If a new motif has a higher impact than the lowest-ranked entry, it replaces it. If not and the row has not been filled, the new motif will be added. The motifs in each row are then re-sorted by their impact scores to retain the most promising candidates.

**Structure-Aware Expression Synthesis.** Once the motif library is updated, new expressions are synthesized by sampling one or more motifs from each row of $\mathbb{M}_f$ until all variables are represented.

These motifs are then assembled into full expressions using randomly selected binary algebraic operators such as $+$, $-$, $\times$, or $\div$, resulting in composite symbolic expressions $\tau'$. The resulting expressions are inserted into a dedicated motif population $P_{\mathcal{M}}$, which co-evolves alongside the main island populations.

**Co-Evolve Motif-Population and Main Populations.** To propagate promising structural components throughout the evolutionary process, high-performing expressions from $P_{\mathcal{M}}$ are periodically injected back into the main populations. This allows strong partial solutions, discovered independently across islands and generations, to be recombined and reused in novel ways, ultimately improving both convergence speed and expression quality.

This structure-aware mechanism complements token-level parameter updates by identifying semantically meaningful subtrees for reuse and recombination. It accelerates symbolic discovery by enabling partial solution reuse and coordinated variable coverage. Motif recombination also improves robustness to local optima by synthesizing expressions from independently validated high-impact components. We visualize this pipeline in Fig. 2, outline the full algorithm in Alg. 1, visualize the full model in Fig. 1, and list hyperparameters in Appendix Tab. A.3.

### 3.4 IMPLEMENTATION DETAILS

SYMMATIKA is a multi-core `C++23` library, compiled with `clang++` and optimized using `-O3` and `-march=native` flags. It leverages `SymEngine` for symbolic computation, `Eigen` for linear algebra and paired-partial derivative operations, and `OpenMP` for parallelization. The full framework is open-sourced to support future work in interpretable modeling and symbolic discovery.

## 4 EXPERIMENTS

For all experiments, SYMMATIKA is instantiated with population sizes of 10,000 individuals per island, followed by truncation to the top 400 for the GP-loop. The number of islands $I$ is set proportional to expression complexity as $I \propto 2d$, where $d$ is the number of variables (capped at $I = 8$ for large SR-problems with five or more variables due to computational considerations). We run each experiment for 1500 generations of GP.

We report: (1) recovery rates (i.e. proportion of exact symbolic expression recoveries, including equivalent constants) on the Nguyen and Feynman benchmark suites (with an emphasis on Nguyen-12 recoveries), (2) error-complexity plots of the Feynman equations (mean proportion of $R^2 > 0.99$ vs. equation complexity) and 57 SRBench Black-box problems with 2-10 (median $R^2$ vs. equation complexity), and (3) implicit-relation discovery in physical systems using Eureqa datasets. We also performed ablation studies to assess the contribution of core components of SYMMATIKA. We report 95% confidence intervals for all compared algorithms and $p$-testing (i.e. two-sample $z$-testing) with algorithms with close performance to SYMMATIKA to showcase the statistical significance of our results. Our goals are to demonstrate the effectiveness of SYMMATIKA on both explicit and implicit SR tasks, highlight cases where it outperforms prior work, and understand how different algorithm modules contribute to performance.

All experiments are run with random seeds and hyperparameters specified in Sec. A.3 on a 2023 Apple M3 Max MacBook Pro with a 14-core CPU, 30-core GPU, and 36GB unified memory. Unlike other neural-guided approaches, SYMMATIKA does not require GPUs nor access to large language models (LLMs). Since SYMMATIKA uses multi-core execution, and many baselines are either single-core or multi-core with unknown runtimes, we report recovery rates (not wall-clock time) for Nguyen and Feynman.

### 4.1 NGUYEN BENCHMARK

Tab. 4 reports recovery rates over 100 independent runs per task. We take reported results from Petersen et al. (2019); Mundhenk et al. (2021) on NGGPPS (Note: PQT is one method of training NGGPPS' RNN), DSR, and Eureqa. SYMMATIKA records an average recovery rate of **96.5%**, outperforming all other methods by statistically significant margins (maximum $p < 10^{-5}$ from $z$-score testing); we also report 95% confidence intervals in Tab. 4. We ran 100 runs with random seeds on Nguyen-12 using PySR and Operon with the exact experiment settings (details in Appendix A.3)

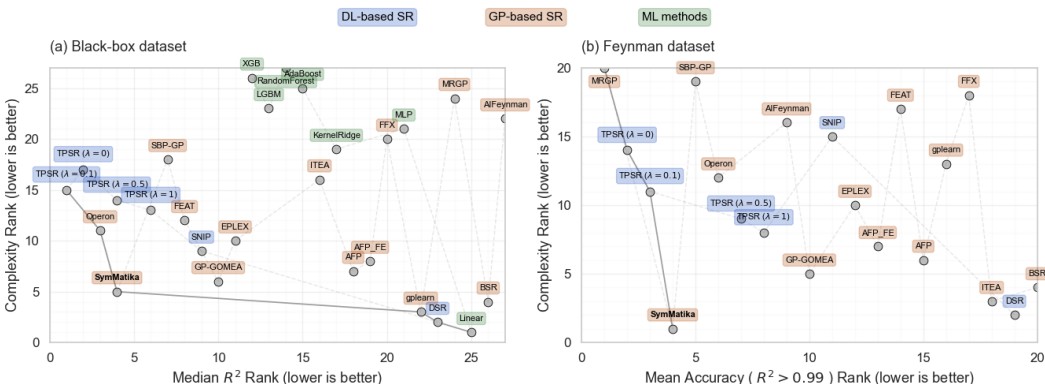

Fig. 3: Error–complexity (rank) trade-off on the Black-box (left) and Feynman (right) datasets.

and reported 0/100 and 2/100 recoveries respectively; given the simpler nature of Nguyen 1-11, we felt it was not necessary to perform testing on problems proven to be easily solvable. Notably NGGPPS (Mundhenk et al., 2021) reports a 12% success rate on a relaxed variant with expanded data range (Nguyen-12⋆), our performance remains significantly stronger on the original Nguyen-12 dataset. These results highlight the advantage of combining structural motif reuse with feedback-driven parameter tuning, especially for long and composite expressions.

## 4.2 FEYNMAN EQUATIONS BENCHMARK

Next, we evaluate on the Feynman benchmark, a widely adopted dataset of 100 symbolic physics expressions derived from the Feynman Lectures. As in prior work, expressions are sampled to form datasets $\mathcal{D} = \{(\mathbf{x}_i, y_i)\}_{i=1}^N$, and success is defined as recovery of the symbolic target, i.e. exact or mathematically equivalent expression. We performed two sets of experiments: (1) recovery rates on 100 Feynman problems outlined in LaSR (Grayeli et al., 2024), and (2) error-complexity plot (mean $R^2 > 0.99$ proportion vs. equation complexity) against SRBench (La Cava et al., 2021) results. We use the same experimental settings as for the Nguyen benchmark (Appendix A.3).

As shown in Tab. 5, SYMMATIKA recovers 73/100 expressions outperforming all methods, including GPlearn (Stephens et al., 2016), DSR (Petersen et al., 2019), uDSR (Landajuela et al., 2022), AIFeynman (Udrescu & Tegmark, 2020), PySR (Cranmer, 2023), and LaSR (Grayeli et al., 2024). LaSR (Grayeli et al., 2024) extends PySR (Cranmer, 2023) with natural language priors from LLMs via a CONCEPTABSTRACTION function, enabling semantic guidance during search. While this approach is orthogonal to ours, our results show that structurally guided recombination and operator adaptation alone are competitive with language-informed strategies. We showcase some sample output in Appendix Sec. A.4.

For the error-complexity plot, SYMMATIKA places impressively on the Pareto front. We report mean $R^2 > 0.99$ proportion of 0.942 and complexity of 13.294 across 6 trials (Fig. 3). We surpass all algorithms on SRBench (e.g. Operon, AIFeynman and SBG-GP to name a few) with maximum $p = 0.044$ — except for MRGP (Arnaldo et al., 2014), which reports mean proportion 0.931 yet complexity 26316, significantly larger than SYMMATIKA. On the Black-box subset, however, we report much better results in both error and complexity then MRGP. Additionally, two versions of TPSR (Shojaee et al., 2023) slightly outperform our mean ($R^2 > 0.99$) accuracy with 0.949 (TPSR[$\lambda = 0.1$]) and 0.952 (TPSR[$\lambda = 0$]) to our 0.942, however our complexity is significantly improved with our 13.294 compared to their 84.42 (TPSR[$\lambda = 0$]) and 57.22 (TPSR[$\lambda = 0.1$]). Additionally, we outperform SNIP (Meidani et al., 2023) at mean $R^2 > 0.99$ proportion 0.876 and complexity 95.709. SYMMATIKA Pareto dominates other SR algorithms (e.g. SNIP, TPSR [$\lambda = 0.5, \lambda = 1$]) and is not dominated by any other algorithm (Fong & Motani); additionally, we lead the front in model complexity. We report all algorithm performances with 95% confidence intervals in Fig. 4 and Tab. 7.

### 4.3 BLACK-BOX BENCHMARK

We evaluated SYMMATIKA on 57 Black-box problems gathered from SRBench used by TPSR (Shojaee et al., 2023) which include synthetic (e.g. synthetic pollen dataset) and real-world (e.g. historical harvest yields in Lake Erie vineyard) datasets and we recorded median $R^2$ and mean equation complexity for the error-complexity plot (Fig. 3). We ran 5 full trials and recorded median $R^2$ of 0.931 and equation complexity of 28.059. We outperform nearly all benchmarked algorithms in median $R^2$, including SNIP at 0.853 (complexity 47.527), and match TPSR [$\lambda = 0.5$] (complexity 82.58), where $\lambda$ is a hyperparameter balancing fitting and complexity reward), and falling slightly behind TPSR [$\lambda = 0$] at 0.938 (complexity 129.85), Operon at 0.937 (complexity 61.74) and TPSR [$\lambda = 0.1$] at 0.945 (complexity 95.71). Our complexity significantly improves on these algorithms by a factor of **3-4.5** for TPSR and a factor of **2** for Operon, and MRGP which reports a much-lower median $R^2$ of 0.502 larger and complexity 9878.172. TPSR records one trial so we cannot report meaningful $p$-testing given limited experimental data, yet we can report that the Operon's outperformance of us is not statistically significant ($p = 0.71138$). SYMMATIKA Pareto-dominates many algorithms, and places strongly on the Pareto-front; notably certain models (e.g. DSR, gplearn) are very simple and so cannot be dominated, yet there accuracy is extremely low.

### 4.4 EUREQA PHYSICAL SYSTEMS

To evaluate implicit symbolic regression, we compared against Eureqa on four time-series physical systems from its supplementary materials, including circular, pendulum, spherical, and double pendulum motions. Unlike the explicit SR benchmarks, these systems do not provide labeled outputs $y$, and recovery involves discovering hidden algebraic constraints $f(\mathbf{x}) = 0$ from the raw dynamics.

SYMMATIKA used the same experimental setup as with previous benchmarks. As Eureqa supports 32-core execution with distributed infrastructure, we report wall-clock convergence times to compare implicit SR performance (Tab. 1). SYMMATIKA recovers the ground-truth implicit expressions outlined in the Eureqa S.O.M. at very improved speeds. We observe ~100× speedups for simpler systems like `circle_1`, and 10× for more complex sys-

Tab. 1: Runtime (s) on Eureqa dataset. SYMMATIKA converges 10×–100× faster.

| System | SYMMATIKA (s) | Eureqa (s) |
|---|---|---|
| circle_1 | 0.15 | 25 |
| pendulum_h_1 | 0.98 | 31 |
| sphere_1 | 3.00 | 320 |
| dbl_pendulum_h_1 | 900.00 | 27000 |

tems like `double_pendulum_h_1`. We believe that these gains stem not only from modern hardware but also from algorithmic improvements, including adaptive operator scheduling, and motif-driven structural reuse.

### 4.5 ABLATION STUDY

To assess the contribution of core components of SYMMATIKA, we perform ablations on the Nguyen benchmark under four configurations: (1) a baseline with standard GP (no parameter tuning or motif reuse), (2) baseline + $\theta$-based parameterized candidate generation, (3) baseline + structural motif library, and (4) the full model with both mechanisms enabled. Each configuration is evaluated across 20 independent runs per task with reported 95% confidence intervals as shown in Tab. 6 in the appendix.

The full model consistently achieves high recovery rates, with improvements observed on nearly every task. Parameterized generation provides consistent boosts across all benchmarks, while motif reuse is especially helpful on longer expressions (e.g., the challenging Nguyen-12 (Sun et al., 2025)). In isolation, the motif mechanism raises the Nguyen-12 recovery rate from 5% to 40%, and further improves to 65% when combined with $\theta$-based parameter adaptation. These results confirm that token-level trends and reusable substructures capture complementary inductive biases that are critical for recovering long or irregular expressions. We discuss limitations in depth in Appendix Sec. A.5.

## 5 CONCLUSIONS

We presented SYMMATIKA, a symbolic regression framework that combines feedback-guided genetic programming with a reusable structural motif library to recover both explicit and implicit expressions from data. Through adaptive operator scheduling, motif-based recombination, and implicit-derivative fitness evaluation, SYMMATIKA outperforms state-of-the-art methods on the Nguyen and Feynman benchmarks and uniquely recovers complex and implicit equations, including Nguyen-12 and Eureqa's physical systems. These results highlight the value of integrating structural priors into symbolic search. Future work will explore robustness to noise and extend the framework with a user-friendly UI to support broader adoption in scientific discovery.

## 6 REPRODUCIBILITY STATEMENT

For reproducibility, we've added a detailed table of model hyperparameters in App. A.3. This table describes each hyperparamater referenced in the main paper — including references to corresponding sections — and includes the exact values used in all our experiments. We also describe the implementation details, including hardware setup, for our experiments in Sec. 3.4. Additionally, we describe the exact experimental setups for the Nguyen (Sec. 4.1), Feynman (Sec. 4.2), Black-box (Sec. 4.3), and Eureqa (Sec. 4.4) datasets.

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

APPENDIX

A.1  SAMPLE OUTPUTS FROM SUBSET OF FEYNMAN EQUATIONS

Tab. 2: Comparison of ground-truth vs. discovered equations.

| Equation Number | Ground Truth Equation | Discovered Equation |
|---|---|---|
| I.18.4 | $r = \frac{m_1 r_1 + m_2 r_2}{m_1 + m_2}$ | $r = \frac{r_1 m_1}{(m_1 + m_2)} + \frac{r_2 m_2}{(m_1 + m_2)}$ |
| II.11.20 | $P_* = \frac{0.333333333333333 E_f p_d^2 n_\rho}{T k_b}$ | $P_* = \frac{n_\rho p_d^2 E_f}{3 k_b T}$ |
| I.37.4 | $I_* = I_1 + I_2 + 2\sqrt{I_1 I_2}\cos(\delta)$ | $I_* = I_1 + I_2 + 2\sqrt{I_1 I_2}\cos(\delta)$ |
| I.24.6 | $E_n = \frac{1}{4}m(\omega^2 + \omega_0^2)x^2$ | $E_n = \frac{1}{2}m(\omega^2 + \omega_0^2)\frac{1}{2}x^2$ |
| III.17.37 | $f = \beta(1 + \alpha\cos(\theta))$ | $f = \beta + \beta\cos(\theta)\alpha$ |
| I.27.6 | $f_f = \frac{d_2}{\left(n + \frac{d_2}{d_1}\right)}$ | $f_f = \frac{1}{\frac{1}{d_1} + \frac{n}{d_2}}$ |
| I.47.23 | $c = \sqrt{\frac{\gamma p r}{\rho}}$ | $c = \sqrt{\frac{\gamma p r}{\rho}}$ |
| I.12.11 | $F = q(E_f + Bv\sin(\theta))$ | $F = qE_f + qBv\sin(\theta)$ |

We demonstrate outputs on a subset of the Feynman Equations and compare ground truth equations to discovered equations. The ground truth equations and discovered equations are syntactially different although mathematically equivalent. These discoveries were found with implementation details consistent with all other experiments. All discovered equations record an MLE loss of $\leq 10^{-16}$.

A.2  SYMMATIKA ALGORITHM

---

**Alg. 1** SYMMATIKA Algorithm

---

**input:** Symbolic regression problem with data $\mathcal{D}$ and relation type $t$
**output:** Best-fitting expression $\tau^\star$

1: Initialize islands $I_1, \ldots, I_m$ with populations $P_1, \ldots, P_m$ and node parameters $\theta_1, \ldots, \theta_m$
2: Initialize fitness trackers $F_1, \ldots, F_m$
3: Define motif library $\mathbb{M}_f$
4: **for** each generation $g$ **do**
5:     **for** each island $I_i$ **do**
6:         $E(I_i) \leftarrow$ evolve island population $P_i$ with $k$ runs of GP
7:         $\tau_u \leftarrow$ update island parameters $\theta_i$
8:         **if** plateau height $h \geq h_{\max}$ **then**
9:             rebuild population $P_i$
10:         **end if**
11:         $\tau_m \leftarrow$ update motif library $\mathbb{M}_f$ with high-performing subexpressions
12:         $M(I_i) \leftarrow$ migrate individuals between islands with growing rate $\alpha_M$
13:     **end for**
14:     $E(\mathbb{M}_f) \leftarrow$ evolve motif population and distribute fittest individuals to island populations
15: **end for**
16: **return** $\tau^\star$

---

A.3  TABULATED SYMMATIKA PARAMETERS

Tab. 3 is a list of SYMMATIKA's experimental settings. We performed each experiment with random seeds.

Tab. 3: Experimental settings of SYMMATIKA.

| Parameters | Explanation |
|---|---|
| $G = 1500$ | number of generations in evolutionary algorithm. |
| $m$ | number of distinct island populations (see Section 4). |
| $k = 200$ | number of iterations of GP per generation (see Fig 1). |
| $t = 0, 1$ | type of SR problem, i.e. *explicit* or *implicit* relation (see Algorithm 1). |
| $\alpha_M = 0.02$ | migration rate of individuals between island populations (see Algorithm 1). |
| $\beta_\theta = (\underbrace{0.2, \ldots, 0.2}_{\text{5 times}}, \underbrace{1.0, \ldots, 1.0}_{\text{13 times}})$ | natural bias term for node type $\theta_i$ (see Section 3.3). |
| $m_i = 0.60, 0.10, 0.30$ | initial genetic operator probabilities (crossover, single-node crossover, mutation) |
| $m_f = 0.05, 0.15, 0.80$ | final genetic operator probabilities (crossover, single-node crossover, mutation) |

## A.4 EXPERIMENTAL RESULTS

Tab. 4: Exact recovery rates (%) on the Nguyen benchmark over 100 runs. SYMMATIKA achieves the highest average performance (96.5%) and recovers Nguyen-12 at significantly higher rates than existing algorithms (61% success). We report 95% confidence intervals.

| Dataset | Expression | SymMatika | NGGPPS | DSR | PQT | Eureqa | Operon | PySR |
|---|---|---|---|---|---|---|---|---|
| Nguyen-1 | $x^3 + x^2 + x$ | 100 | 100 | 100 | 100 | 100 | — | — |
| Nguyen-2 | $x^4 + x^3 + x^2 + x$ | 100 | 100 | 100 | 99 | 100 | — | — |
| Nguyen-3 | $x^5 + x^4 + x^3 + x^2 + x$ | 100 | 100 | 100 | 86 | 95 | — | — |
| Nguyen-4 | $x^6 + x^5 + x^4 + x^3 + x^2 + x$ | 100 | 100 | 100 | 93 | 70 | — | — |
| Nguyen-5 | $\sin(x^2)\cos(x) - 1$ | 100 | 100 | 72 | 73 | 73 | — | — |
| Nguyen-6 | $\sin(x) + \sin(x + x^2)$ | 100 | 100 | 100 | 98 | 100 | — | — |
| Nguyen-7 | $\log(x + 1) + \log(x^2 + 1)$ | 98 | 97 | 35 | 41 | 85 | — | — |
| Nguyen-8 | $\sqrt{x}$ | 100 | 100 | 96 | 21 | 0 | — | — |
| Nguyen-9 | $\sin(x) + \sin(y^2)$ | 100 | 100 | 100 | 100 | 100 | — | — |
| Nguyen-10 | $2\sin(x)\cos(y)$ | 100 | 100 | 100 | 91 | 64 | — | — |
| Nguyen-11 | $x^y$ | 100 | 100 | 100 | 100 | 100 | — | — |
| Nguyen-12 | $x^4 - x^3 + \frac{1}{2}y^2 - y$ | **61** | 0 | 0 | 0 | 0 | 2 | 0 |
| **Average** | | **96.5 ±0.608** | 91.4 ±1.560 | 83.6 ±1.829 | 75.2 ±1.885 | 73.9 ±2.001 | — | — |

Tab. 5: Exact recovery counts on the Feynman benchmark (100 tasks). SYMMATIKA recovers 73 expressions, outperforming all other algorithms.

| GPlearn | AFP-FE | DSR | uDSR | AIFeynman | PySR | LaSR | SYMMATIKA |
|---|---|---|---|---|---|---|---|
| 20/100 | 26/100 | 23/100 | 40/100 | 38/100 | 59/100 | 72/100 | **73/100** |

To showcase some sample output, we run SYMMATIKA on a sample equation from the Feynman Lectures on Physics (Feynman et al., 2015), the relativistic Doppler shift formula $\omega = \frac{1 + \frac{v}{c}}{\sqrt{1 - \frac{v^2}{c^2}}}\omega_0$.

After running SYMMATIKA on its dataset, it returns the expression $\omega = \sqrt{\frac{c+v}{(c-v)}}\omega_0$ with an MLE loss of $-3.10862 \times 10^{-16}$, and the discovered expression is mathematically equivalent to the ground truth expression, albeit syntactically slightly different. We provide more outputs on a subset of sampled Feynman Equations in Appendix Sec. A.1.

## A.5 Ablations & Limitations

Tab. 6: Recovery rates for individual benchmark problems with model ablations across 20 independent trials per problem. 95% confidence intervals are obtained from the recovery rates across all 12 Nguyen problems for each ablation.

| Problem | Recovery rate (%) | | | |
| --- | --- | --- | --- | --- |
| | **Baseline** | $\theta$ **Generation** | **Motif Library** | **Full Model** |
| Nguyen-1 | 100 | 100 | 100 | 100 |
| Nguyen-2 | 100 | 100 | 100 | 100 |
| Nguyen-3 | 100 | 100 | 100 | 100 |
| Nguyen-4 | 90 | 95 | 100 | 100 |
| Nguyen-5 | 95 | 100 | 100 | 100 |
| Nguyen-6 | 85 | 90 | 95 | 100 |
| Nguyen-7 | 60 | 80 | 85 | 95 |
| Nguyen-8 | 100 | 100 | 100 | 100 |
| Nguyen-9 | 100 | 100 | 100 | 100 |
| Nguyen-10 | 100 | 100 | 100 | 100 |
| Nguyen-11 | 100 | 100 | 100 | 100 |
| Nguyen-12 | 5 | 20 | 40 | 65 |
| **Nguyen average** | 86.3 $\pm15.92$ | 90.4 $\pm13.02$ | 93.3 $\pm9.82$ | 96.7 $\pm5.70$ |

The primary limitation in our approach is our results on the Feynman Equations. Although we report results outperforming or performing on par with leading models, we are still unable to solve a subset of 27 problems. These problems generally involve long expressions with complicated structures, such as $n = \frac{n_0}{\exp(\mu_m B/(k_b T)) + \exp(-\mu_m B/(k_b T))}$, problem II.35.18 in the Feynman Equations (Udrescu & Tegmark, 2020). Other unsolved problems include large arguments in unary operators (e.g. $x = \sqrt{x_1^2 + x_2^2 - 2x_1 x_2 \cos(\theta_1 - \theta_2)}$). These structures contain very few distinct subtrees and therefore provide limited opportunities for meaningful recombination. Even so, we retain state-of-the-art performance on the Feynman Equations. To attempt further discovery of Feynman equations, we will conduct experiments in an improved hardware setup with more CPU cores to evolve more island populations in parallel with larger populations and for longer. Additionally, we will further investigate methods of discovering implicit relations in higher-dimensional data (i.e. $\geq 5$ variables) using our implicit derivative fitness metric.

Tab. 7: Black-box vs. Feynman benchmark predictive accuracy and model complexity performances.

| Black-box | | |
|---|---|---|
| Algorithm | Median $R^2$ | Complexity |
| TPSR ($\lambda$=0.1) | 0.945 | 95.710 |
| TPSR ($\lambda$=0) | 0.938 | 129.850 |
| Operon | 0.938 | 61.742 |
| **SymMatika** | 0.931 | 28.059 |
| TPSR ($\lambda$=0.5) | 0.931 | 82.580 |
| TPSR ($\lambda$=1) | 0.924 | 79.430 |
| SBP-GP | 0.916 | 616.735 |
| FEAT | 0.906 | 72.664 |
| SNIP | 0.853 | 47.527 |
| GP-GOMEA | 0.836 | 32.095 |
| EPLEX | 0.836 | 53.147 |
| XGB | 0.817 | 19133.300 |
| LGBM | 0.794 | 4822.224 |
| RandomForest | 0.753 | 1155301.000 |
| AdaBoost | 0.747 | 9892.822 |
| ITEA | 0.736 | 103.935 |
| KernelRidge | 0.729 | 1170.091 |
| AFP | 0.710 | 36.262 |
| AFP_FE | 0.702 | 36.864 |
| FFX | 0.665 | 1429.411 |
| MLP | 0.638 | 1794.727 |
| gplearn | 0.638 | 22.744 |
| DSR | 0.600 | 10.556 |
| MRGP | 0.502 | 9878.172 |
| Linear | 0.342 | 6.964 |
| BSR | 0.184 | 23.356 |
| AIFeynman | -0.608 | 1989.177 |

| Feynman | | |
|---|---|---|
| Algorithm | Mean $R^2 > 0.99$ | Complexity |
| MRGP | 0.977 | 3669.357 |
| TPSR ($\lambda$=0) | 0.952 | 84.420 |
| TPSR ($\lambda$=0.1) | 0.949 | 57.220 |
| **SymMatika** | 0.942 | 13.294 |
| SBP-GP | 0.941 | 489.416 |
| Operon | 0.940 | 69.875 |
| AIFeynman | 0.885 | 124.478 |
| SNIP | 0.882 | 31.630 |
| GP-GOMEA | 0.880 | 34.571 |
| EPLEX | 0.780 | 52.945 |
| AFP_FE | 0.733 | 39.966 |
| FEAT | 0.616 | 205.305 |
| AFP | 0.568 | 36.869 |
| gplearn | 0.504 | 72.427 |
| FFX | 0.425 | 271.693 |
| ITEA | 0.405 | 21.098 |
| DSR | 0.358 | 14.857 |
| BSR | 0.222 | 25.500 |

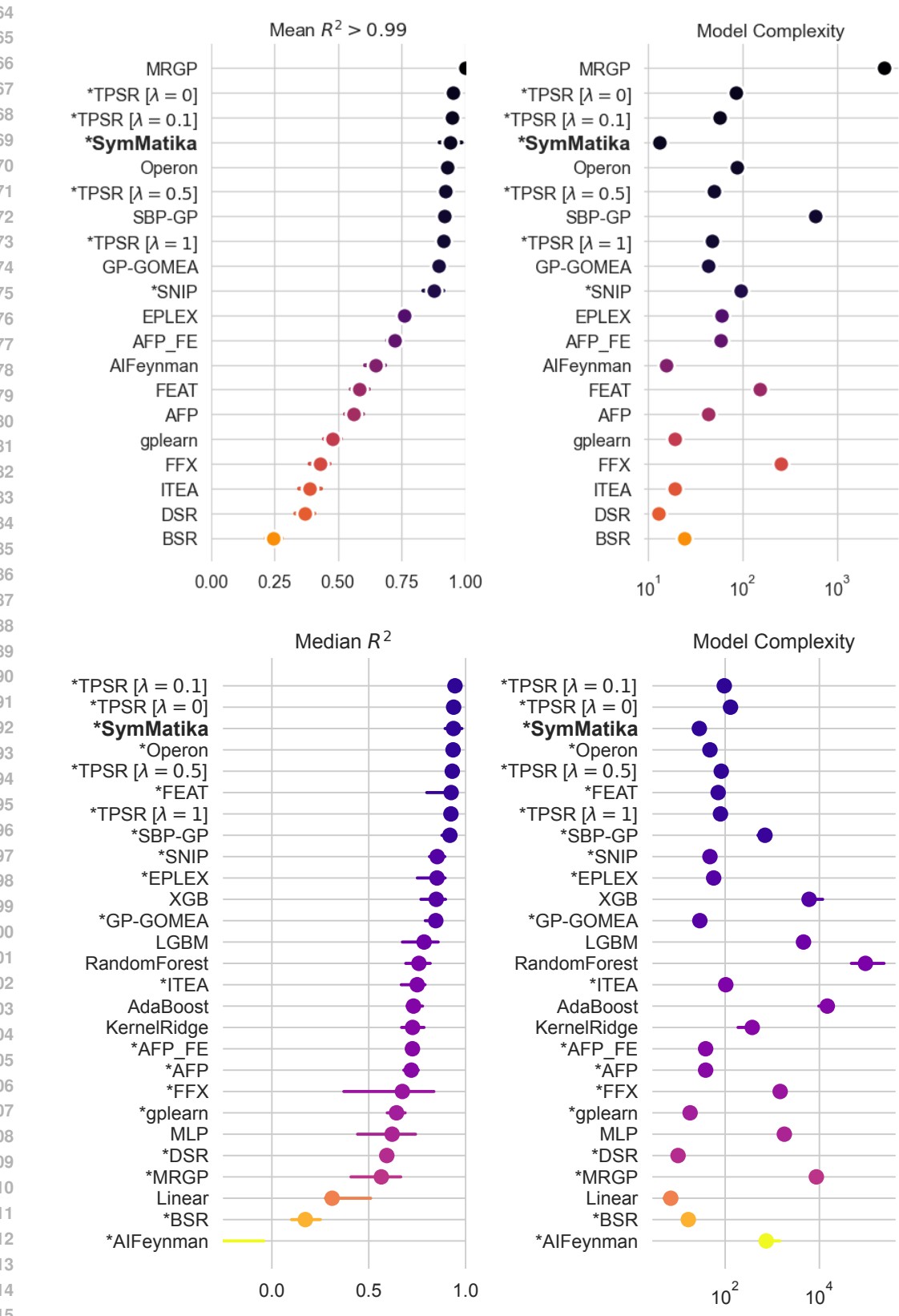

Fig. 4: 95% confidence intervals for (top) Feynman equations and (bottom) Black-box benchmarks. "*" refers to SR methods for *Black-box* datasets.

