# OpenReview forum: "SymMatika: Structure-Aware Symbolic Discovery"
_ICLR.cc/2026/Conference — Submitted to ICLR 2026_

### Official Review · Reviewer_C1gH · 2025-10-20

**Soundness:** 2
**Presentation:** 3
**Contribution:** 2
**Rating:** 4
**Confidence:** 4

**Summary:**

This paper proposes SYMMATIKA, a symbolic regression framework that combines multi-island genetic programming and a reusable symbol library to accelerate search, supporting both explicit (y=f(x)) and implicit (F(x,y)=0) regression tasks. Experimental results demonstrate its superiority over existing methods on benchmarks, particularly in recovery rate and computational efficiency.

**Strengths:**

The proposed method can simultaneously support both explicit and implicit regression tasks, and demonstrates superior performance over existing approaches across multiple benchmark tests.

**Weaknesses:**

This paper appears to be more of a combination and fine-tuning of existing proposed methods.
For instance, the paper introduces a Boltzmann selection mechanism, which has already been in use in the Evolutionary Algorithms  field for over 20 years. The multi-island evolution mechanism is also a very classic approach. The paper proposes a feedback-based operator scheduling mechanism, but the design of this mechanism seems ad-hoc, lacking sufficient explanation and theoretical depth. Search mechanisms based on a reusable library have also been proposed by scholars in the Genetic Programming field long ago.
Furthermore, the review of relevant literature is insufficient. For example, the discussion on Implicit Regression only involves a single paper from 2009, overlooking new methods proposed in recent years. For implicit regression, the experimental section only analyzes the method on four simple test functions and is compared against only one other algorithm, making the results unconvincing.

**Questions:**

1. The paper combines several established techniques (e.g., Boltzmann selection, multi-island evolution, reusable library search). What is the theoretical foundation that unifies these techniques into a novel methodology?
2. The experiments for implicit regression focus on four test functions and one baseline, while related work discussion omits recent advances in implicit regression. How does the proposed method explicitly improve upon  recent implicit regression techniques, and what justifies the choice of the baseline comparison algorithm?

---

> ### Author Response · Authors · 2025-11-22
>
> We thank the reviewer for the thoughtful comments. We appreciate the reviewer’s recognition of our unified support of explicit and implicit regression and state-of-the-art experimental results on multiple benchmark tests. We would like to address all of your concerns and questions below with point responses.
>
> **“This paper appears to be more of a combination and fine-tuning of existing proposed methods … Boltzmann selection mechanism … multi-island evolution mechanism”**
>
> Our main contribution is adapting the search procedure based on both positive parameter and structural-level trends in symbolic equations, notably through our integration of a motif library for structural subexpression reuse. Although we use features like Boltzmann selection and multi-island evolution, they are not this paper’s main contribution. Moreover, SymMatika also supports both explicit and implicit equation discovery, which is unique among existing SR algorithms.
>
> **“The paper proposes a feedback-based operator scheduling mechanism, but the design of this mechanism seems ad-hoc, lacking sufficient explanation and theoretical depth”**
>
> We would like to clarify that our operator-scheduling mechanism is not ad-hoc, but a principled heuristic grounded in GP search dynamics. As described in lines 233–252, coarse-grained operators are most effective early in evolution, while fine-grained edits help once populations plateau. Our schedule directly encodes these dynamics by adjusting operator probabilities based on measurable evolutionary signals in relative fitness improvements (formula at line ~244) and plateau duration *h*. The bounds $m_i$​ and $m_f$​ ensure stability and prevent uncontrolled drift [lines 257-259].
>
> **“Search mechanisms based on a reusable library have also been proposed by scholars in the Genetic Programming field long ago”**
>
> We would like to clarify that, although similar mechanisms have been previously proposed, it is a feature that is absent in modern GP-based symbolic regression. Previous GP algorithms like LaSR use LLMs to update abstract concept libraries in evolution, but none update libraries of high-impact symbolic subexpressions.
>
> **“The paper combines several established techniques … What is the theoretical foundation that unifies these techniques into a novel methodology?”**
>
> Our key idea is that the search distribution over symbolic expressions should be continuously adapted using information extracted from the evolutionary process at two complementary levels: parameter-level trends and structural-level advantageous subtrees. Although we integrate established SR features, our novelty and foundation lies in this principle of continuous adaptation based on positive parameter-level and structural trends in evolution. Moreover, SymMatika supports both explicit and implicit regression, while past methods support only one of them.
>
> **“... discussion on Implicit Regression only involves a single paper from 2009, overlooking new methods proposed in recent years … the experimental section only analyzes the method on four simple test function and is compared against only one other algorithm” “How does the proposed method explicitly improve upon recent implicit regression techniques, and what justifies the choice of the baseline comparison algorithm?”**
>
> Though released early and outdated in some respects, the benchmarked problem suite features time-series physical systems with no known priors; certain implicit regression benchmarks, like in PIE, require reordering explicit regression problems, limiting their capability of implicit regression in unknown and chaotic physical systems. Our results have shown that SymMatika has provided strong results in these settings such as real-world chaotic physical systems. Most other recent advancements have focused on explicit regression as shown in our experiments. SymMatika offers both explicit and implicit regression with state-of-the-art performance, which makes our contribution unique. Though Eureqa provides strong performance, it is no longer available and has remained close sourced since beginning. SymMatika will be completely open sourced. We believe this is another contribution to the community to study both explicit and implicit regression tasks. SymMatika offers a unified method for both families of tasks.

---

> > ### Author Response · Authors · 2025-12-03
> > **Author Response Summary**
> >
> > We thank the reviewer again for the thoughtful comments, and appreciate their recognition of unique support for dual explicit and implicit regression and state-of-the-art performance on leading benchmarks. In our response and revisions to the paper, we clarified that our main contribution is adapting the search procedure based on both positive parameter and structural-level trends in symbolic equations. While we use features like Boltzmann selection and multi-island evolution, they are strictly complementary with proven advantages in GP-based SR and are not this paper’s main contribution. We explicitly noted our support for both explicit and implicit regression cases, a rare feature among modern SR algorithms. We clarified that our operator-scheduling mechanism is grounded in GP search dynamics. While similar evolutionary mechanisms have been proposed, we clarified that our search mechanisms have been absent in modern SR algorithms and that no existing algorithms update libraries of high-impact symbolic subexpressions. We explained that our main contribution is continuously adapting the search distribution over symbolic expressions using information extracted from both parameter-level trends and structural-level advantageous subtrees, and that this forms a fully novel methodology. We justified our choice of implicit regression benchmark due to it including time-series physical systems with no known priors, while other baselines, like PIE, require reordering explicit regression problems which limit their capability of implicit regression in unknown and chaotic physical systems, while SymMatika can fully handle these cases with impressive performance. We also noted that SymMatika will be completely open-sourced, unlike Eureqa which has remained close sourced since beginning. This is another contribution to the community to study both explicit and implicit regression tasks, where SymMatika offers a unified method for both families of tasks.

---

### Official Review · Reviewer_yMEA · 2025-10-28

**Soundness:** 3
**Presentation:** 2
**Contribution:** 3
**Rating:** 4
**Confidence:** 3

**Summary:**

This paper proposes SYMMATIKA, a structure-aware symbolic regression (SR) framework that can discover both explicit relations and implicit relations.  SYMMATIKA integrates two main innovations: 1) Feedback-driven multi-island genetic programming, which adaptively tunes mutation, crossover, and selection rates based on evolutionary progress. 2) A reusable motif library, inspired by biological sequence motifs, to identify and reuse high-impact symbolic subexpressions for faster convergence. The method achieves state-of-the-art recovery rates on Nguyen, Feynman, SRBench, and Eureqa benchmarks.  Notably, it recovers 96.5% of Nguyen tasks (including 61% success on Nguyen-12, compared to 2% for prior methods), and converges 10–100× faster than Eureqa on implicit physical systems.  The system is fully implemented in optimized C++ and requires only multi-core CPUs.

**Strengths:**

1.Novel integration of motif-level structural reuse in symbolic regression.

2.Feedback-driven adaptive operator scheduling for efficient search.

3.Supports both explicit and implicit relation discovery.

4.Strong empirical results across multiple standard benchmarks.

5.No reliance on deep neural networks or GPUs, making it computationally efficient and accessible.

6.Unified symbolic regression framework for both explicit and implicit relations.

**Weaknesses:**

1. **Limited theoretical analysis of convergence and motif importance.**

   The paper does not provide a formal convergence argument for the co-evolution of motifs and populations (Sec. 3.3–3.4). Motif impact $I(\tau') = L(\tau) - L(\tau - \tau')$ is introduced heuristically (line 302 – 310) without theoretical justification that this local fitness differential leads to global improvement.  While empirically effective, there is no analysis of stability (e.g., whether motif reuse can cause premature convergence). Related to convergence theory in GP (e.g., Whitley 1999, Doerr 2019), SYMMATIKA’s dynamics remain descriptive rather than proven.

2. **Evaluation lacks noisy and high-dimensional datasets (≥5 variables).**

   Experiments (Sec. 4, Tables 4–5) primarily involve low-dimensional benchmarks (Nguyen: ≤2 vars; Feynman: ≤4 vars).  The authors explicitly note in Appendix A.5 (L791 – L809) that performance deteriorates for equations with ≥5 variables, acknowledging scalability issues.  No robustness study under observational noise was performed, which limits the method’s applicability to real scientific data.

3. **Some Feynman equations remain unsolved; scalability is limited.**

   Appendix A.5 reports 27 unsolved Feynman problems, particularly those with long nested unary operators or composite exponentials (e.g., problem II.35.18).  The authors attribute this to limited population size and CPU-only hardware, but it also indicates that motif reuse and feedback adaptation do not fully handle high-complexity functional structures.  This limitation suggests that the system still struggles with extremely deep symbolic trees.

4. **Results mostly deterministic; stochastic robustness not studied.**

   All experiments (Sec. 4.1–4.4) are run with fixed hyperparameters and deterministic seeds.  There is no variance or sensitivity analysis across random seeds, mutation rates, or population sizes.  As a result, the general stability of convergence and reproducibility under stochastic perturbations remain uncertain.

**Questions:**

See the weaknesses.

---

> ### Author Response · Authors · 2025-11-22
>
> We thank the reviewer for the thoughtful comments. We appreciate the reviewer’s recognition of our unified approach to explicit and implicit regression and our novel integration of structural subexpression reuse and feedback-based operator scheduling. We would like to address all of your concerns and questions below with point responses.
>
> **“The authors explicitly note in Appendix A.5 (L791 – L809) that performance deteriorates for equations with ≥5 variables”**
>
> We would like to clarify that we did not make the claim that the performance deterioration for equations with ≥5 variables. Instead, we stated “additionally, we will further investigate methods of discovering **implicit relations** in higher-dimensional data (i.e. ≥ 5 variables),” as available standard test data from Eureqa physical systems contain only up to 4 variables. In fact, we demonstrated strong performance on 2-10 variable equations in the Black-box results; we have added a quick note of variable size of the Black-box equations [lines 357] to resolve this confusion.
>
> **“No robustness study under observational noise was performed, which limits the method’s applicability to real scientific data”**
>
> We would like to clarify that we have tested on real scientific data with varying levels of noise (e.g. historical harvest yields in Lake Erie vineyard) in the Black-box benchmark problems [lines 355-357], where we reported leading rank than other baseline algorithms on the error-complexity Pareto front which demonstrates SymMatika’s applicability to real scientific data. We also clarified this point in the revised paper.
>
> **“The paper does not provide a formal convergence argument for the co-evolution of motifs and populations … without theoretical justification that this local fitness differential leads to global improvement”**
>
> SymMatika periodically injects high-performing motif expressions back into the populations [lines 329-331]. These expressions are added into the populations iff they improve the average-fitness of top-performers (i.e. top-M performers). Therefore, SymMatika enjoys similar convergence to global improvements as evolutionary-based symbolic regression methods. Notably, other SR algorithms like PySR, which employs temperature-based evolution to alternate between high and low temperature search phases, similarly use evolutionary strategies to optimize global convergence. SymMatika shares the same convergence with these algorithms.
>
> **“Appendix A.5 reports 27 unsolved Feynman problems, particularly those with long nested unary operators or composite exponentials … it also indicates that motif reuse and feedback adaptation do not fully handle high-complexity functional structures … the system still struggles with extremely deep symbolic trees”**
>
> Although we struggle to solve a subset of Feynman problems, we retain state-of-the-art performance. Expressions with long unary chains or composite exponentials are difficult for all GP-based SR methods, as these structures contain very few distinct subtrees and therefore provide limited opportunities for meaningful recombination. This is the primary reason these expressions form the majority of our remaining unsolved Feynman cases (Appendix A.5). We have clarified this in the revision.
> Importantly, this limitation is restricted to unary-heavy expressions rather than to “deep” trees in general. SymMatika successfully handles high-complexity, multi-variable SR problems in other regimes, as reflected by its strong Pareto-front placement on the Black-box benchmark.
>
> **“All experiments (Sec. 4.1–4.4) are run with fixed hyperparameters and deterministic seeds … There is no variance or sensitivity analysis across random seeds, mutation rates, or population sizes”**
>
> All experiments are run with multiple random seeds [line 701, App. A.3], though we can make this clearer for readers. Mutation rates are self-adjusting (lines 108-110). By conducting experiments on multiple random seeds, our experiments both follow standard practice in evaluating symbolic regression systems and report overall and general performance with reproducible outcomes for future comparisons.

---

> > ### Comment · Reviewer_yMEA · 2025-11-24
> >
> > The authors' response has addressed most of my concerns; however, I will maintain my original score.

---

> > > ### Author Response · Authors · 2025-11-24
> > >
> > > Thank you for noting that our revisions addressed most of your earlier concerns. We appreciate your efforts and time to review our paper. We respect your decision to maintain the original score. If there are specific areas you believe still limit the score, we would be grateful for any further suggestions or questions.

---

> > > > ### Author Response · Authors · 2025-12-03
> > > > **Author Response Summary**
> > > >
> > > > We thank the reviewer again for the thoughtful comments and appreciate their recognition of SymMatika’s novel integration of structural subexpression reuse and feedback-based operator scheduling. In our response and revisions to the paper, we clarified confusions about claims made in the paper, and provided explicit evidence of our true claims of strong performance scaling to high-dimensional problems (e.g. 2-10 variable datasets for the Black-box benchmark). We clarified that SymMatika was tested on real scientific data (e.g. historical harvest yields in Lake Erie vineyard as one example of several) with varying levels of noise. Additionally, we further clarified that we did report leading rank on the error-complexity Pareto front, demonstrating our applicability to real scientific data. We highlighted that co-evolution of motifs and feedback-based operator scheduling are effective methods to improve convergence, citing relevant parts of the paper explaining methodology. We justified model components, like our injection of high-performing motif expressions back into the populations, and explained that other GP-based SR algorithms like PySR employ features like temperature-based evolution to optimize global convergence, while demonstrating lesser performance on leading benchmarks than SymMatika. Although we struggled to solve a subset of Feynman problems, we retained state-of-the-art performance and pointed out that how expressions with long unary chains or composite exponentials are still difficult for all GP-based SR methods, which we clarified explicitly in our revised paper. We emphasized this limitation is restricted to unary-heavy expressions rather than “deep” trees in general, as SymMatika successfully handles high-complexity, multi-variable SR problems in other regimes, as reflected by its strong Pareto-front placement on the Black-box benchmark. We clarified that all experiments are in fact run with multiple random seeds and that mutation rates were self-adjusting, and these practices allowed for reproducible outcomes for future comparisons and reporting overall and general performance.

---

### Official Review · Reviewer_FFxE · 2025-10-30

**Soundness:** 3
**Presentation:** 3
**Contribution:** 3
**Rating:** 6
**Confidence:** 5

**Summary:**

The work introduces SymMatika, a symbolic regression algorithm that integrates multi-island genetic programming with a motif library for structural reuse and feedback-driven operator scheduling.

**Strengths:**

Conceptual novelty in motif library that measures impact via ablation and re-injects high-impact subexpressions.

Suitable benchmark datasets and benchmark algorithms are evaluated, with appropriate ablation study (albeit a very short one on a single benchmark, i.e., only Nguyen).

Tackles both explicit and implicit relationships, in which the latter is impactful but has not been addressed much in existing literature.

Organization and flow of paper is well-designed.

**Weaknesses:**

It is hard to see what the gap in performance for SymMatika with its adjacent algorithms. For example, in Fig. 3, it seems that Operon is better than SymMatika in terms of R^2, but maybe the difference is only less than 0.01 R^2. It is not possible for the reader to determine the difference as it is reported now. [1] has also proved that adding or removing algorithms that are not on the Pareto front can paradoxically cause the set of Pareto-optimal algorithms to change when aggregating ranks. Thus, I would recommend to analyze Pareto-optimality by using the actual metric, instead of the rank of the metric.

[1] Fong, Kei Sen, and Mehul Motani. "Pareto-Optimal Fronts for Benchmarking Symbolic Regression Algorithms.", In ICML’25

No confidence intervals or error bars, but there is an indication of z-score testing. Can the paper describe the statistical test procedure in more details? I had assumed Wilcoxon ranked sign test would be the de facto test, but I would like to know more details about the full testing procedure used. Also, can the paper include the results of SymMatika against the other methods as well, I can only find the results for Operon vs SymMatika. This is easy to resolve, so in the recommendation score, I am assuming that the confidence intervals/error bars and statistical test results for the other comparisons will be added.

I can’t seem to find SNIP [2] in the left of Figure 3. Why is this so?

[2] Meidani, Kazem, et al. "SNIP: Bridging mathematical symbolic and numeric realms with unified pre-training.", In ICLR’24

I also cannot find the work [2] cited in the references list, despite being used as a baseline in the right of Figure 3. I have not checked for all the other algorithms, SNIP only came to mind because it was used inconsistently, as mentioned above. Can the paper make sure that all algorithms used in the paper are cited?

Please provide a clear, self-contained, reproducible definition of equation "recovery" that is used in this paper. If constant matching is involved, please explain how 2 constants are determined to be "equal".

**Questions:**

Please address the questions in the weakness section, thanks.

---

> ### Author Response · Authors · 2025-11-22
>
> We thank the reviewer for the thoughtful comments. We appreciate the reviewer’s recognition of our novel motif library, support for explicit and implicit regression, and suitable choice of benchmarks. We would like to address all of your concerns and questions below with point responses.
>
> **“It is hard to see what the gap in performance for SymMatika with its adjacent algorithms. For example, in Fig. 3, it seems that Operon is better than SymMatika in terms of $R^2$, but maybe the difference is only less than 0.01 $R^2$. It is not possible for the reader to determine the difference as it is reported now.”**
>
> We would like to clarify that we have also stated Operon’s $R^2$ performance among other SR algorithms [lines 442-444] and compared with SymMatika. We acknowledge that this is a fair critique and we have added a figure in the Appendix (Fig.4 in Appendix) and a table (Tab.7 in Appendix) with numerical values on $R^2$ and complexity for each algorithm for detailed numerical comparison in addition to Fig. 3 and our descriptions in lines 442-444.
>
> **“I would recommend to analyze Pareto-optimality by using the actual metric, instead of the rank of the metric”**
>
> We have added descriptions of SymMatika’s relative Pareto domination on both Feynman and Black-box benchmarks. Finding absolute Pareto-optimal (APO) solutions proposed by Fong et al. would require computing their proposed algorithm across well over a hundred datasets. Due to the limited time for rebuttal, we will not be able to complete them on time. However, we would like to note that our relative Pareto domination of SymMatika has shown sufficient evidence to prove the effectiveness of our method.
>
> **“Can the paper describe the statistical test procedure in more detail … this is easy to resolve, so in the recommendation score, I am assuming that the confidence intervals/error bars and statistical test results for the other comparisons will be added”**
>
> We have added a brief explanation of statistical testing procedure in Experiments, before analyzing experimental results. We have added confidence intervals/error bars and statistical test results for the other comparisons as requested by the reviewer.
>
> **“I can’t seem to find SNIP [2] in the left of Figure 3. Why is this so? … Can the paper make sure that all algorithms used in the paper are cited?”**
>
> It seems that the SNIP algorithm was clipped from the plot when rendered, we have added it back. All algorithms used in the paper have been cited in our revised paper.
>
> **“Please provide a clear, self-contained, reproducible definition of equation "recovery" that is used in this paper. If constant matching is involved, please explain how 2 constants are determined to be "equal".”**
>
> We define recovery rates as exact recovery of the symbolic target [407], but we’ve made this clearer in the beginning of Experiments [lines 354-362]. Constants are “equal” when there is no difference in symbolic expression MLE loss for different constants, and we can make this clearer as well.

---

> > ### Author Response · Authors · 2025-12-03
> > **Author Response Summary**
> >
> > We thank the reviewer again for the thoughtful comments, and appreciate their recognition of our novel motif library, dual support of explicit and implicit regression and suitable choice of benchmarks. In our response and revision of the paper, as requested by the reviewer, we reported all numerical values for $R^2$ and complexity as well as relevant 95% confidence intervals and error bars and we explicitly described in detail the statistical testing procedures we took. We added descriptions of SymMatika’s relative Pareto domination on both Feynman and Black-box benchmarks noting our superiority over existing methods, and explained our results and evaluation metrics already adopted the standard practices and proved the effectiveness of our method. All algorithms have been properly cited and the figures are updated with these revisions, along with improving our existing definition of “recovery” to improve reader clarity.

---

### Official Review · Reviewer_Ljt9 · 2025-10-31

**Soundness:** 3
**Presentation:** 3
**Contribution:** 2
**Rating:** 6
**Confidence:** 3

**Summary:**

This work proposed a novel symbolic regression framework-SYMMATIKA that combines feedback-driven genetic programming with a reusable structural motif library to discover both explicit and implicit mathematical expressions from data. By leveraging adaptive operator scheduling, motif-based recombination, and implicit-derivative fitness evaluation, SYMMATIKA achieves state-of-the-art recovery rates on standard benchmarks like Nguyen and Feynman equations, significantly outperforming existing methods in accuracy, convergence speed, and model complexity.

**Strengths:**

1. SYMMATIKA supports both explicit and implicit relation discovery, making it applicable to a wide range of scientific problems.

2.The introduction of a motif library enables the identification and recombination of high-impact substructures, accelerating convergence and improving robustness to local optima.

**Weaknesses:**

However, the framework has limitations, including its inability to solve certain complex Feynman equations and a lack of experiments on real-world applications.

While the results are promising, some comparisons lack statistical significance due to limited experimental data.

Furthermore, the novelty of SYMMATIKA is somewhat constrained as it builds upon  implicit-derivative metrics, and its structural motif reuse, while innovative, is inspired by established concepts from biological sequence analysis.

**Questions:**

What specific improvements to the algorithm or hardware setup do you believe could help tackle the subset of unsolved Feynman equations? ​ Are there plans to incorporate additional techniques, such as deep learning or domain-specific priors, to address these challenges?

How does SYMMATIKA's structural motif reuse compare to neural-guided symbolic regression methods in terms of scalability and interpretability? Could the framework benefit from integrating neural network components for further optimization?


While SYMMATIKA demonstrates impressive speed improvements over Eureqa, how does it compare in terms of computational resource requirements with other modern SR frameworks like PySR or TPSR?

---

> ### Author Response · Authors · 2025-11-22
>
> **Response 1/2**
>
> We thank the reviewer for the thoughtful comments. We would like to address all of your concerns and questions below with point responses. We appreciate the reviewer’s recognition of our support for explicit and implicit regression, and our motif library’s efficacy in accelerated convergence.
>
> **“Framework has limitations … lack of experiments on real-world applications"**
>
> We would like to clarify that we did perform tests on both synthetic and real-world datasets [lines 436-437] with the Black-box benchmark suite. This includes data on historical harvest yields in Lake Erie vineyard. We have highlighted this in our revised paper [lines 436-437]
>
> **“Some comparisons lack statistical significance due to limited experimental data”**
>
> This is because certain baseline algorithms did not report enough trials to perform rigorous statistical testing against our own algorithm. We conducted statistical tests for all algorithms with sufficient experimental data and mentioned algorithms which we couldn’t provide the same tests for given lack of data from such algorithms [e.g. lines 444-446].
>
> **“The novelty of SymMatika is somewhat constrained…builds upon implicit-derivative metrics”**
>
> SymMatika is novel in that it supports symbolic regression for **both** explicit-mappings and implicit-relations, where most leading SR algorithms support one. Importantly, our approach’s novelty also lies in its continuous adaptation of the search distribution over symbolic expressions using both parameter-level trends and structural-level advantageous subtrees. This method is unique in the field and powerful based on our experimental result success, and our ablations with and without our novel structural reuse library.
>
> **“Structural motif reuse, while innovative, is inspired by established concepts from biological sequence analysis”**
>
> We agree that this concept is inspired by biological sequence analysis, since we have already mentioned this inspiration in our paper [lines 294-297]. However, we would like to highlight that this does not hurt our contributions, as many computer-realized algorithms are inspired from biology and nature. Instead, being able to ground these high-level inspirations and demonstrate strong results is a solid contribution. Moreover, though SymMatika identifies high-impact subexpressions in top-performing candidates, the methods of determining high-impact are unique even in biological sequence analysis. Our algorithm determines high-impact by measuring fitness contributions of subtrees in larger symbolic expression trees. This is a new adaptation in symbolic regression and GP-based search, with verifiable efficacy demonstrated by our experimental results across leading benchmarks and ablation studies.

---

> ### Author Response · Authors · 2025-11-22
>
> **Response 2/2**
>
> **“What specific improvements to the algorithm or hardware setup do you believe could help tackle the subset of unsolved Feynman equations”**
>
> Algorithmically, producing composite motif expressions as arguments for unary operators could improve convergence to correct expressions with long nested unary operators or composite exponentials. This is a natural extension of our structural-reuse framework. However, the motif mechanism introduced in this paper is designed specifically to evaluate the contribution of high-impact subexpressions and their recombination with algebraic operators. Extending the system to support compositional embeddings inside unary operators requires an expanded motif-library and impact-estimation pipeline, effectively a new architectural component. We therefore treated this as beyond the scope of the present work, which focuses on demonstrating the benefits of structural reuse and feedback-driven operator scheduling within the existing GP framework.
> Hardware-wise, running on a computer with more CPU cores would allow for larger algorithm parallelization, including more populations and faster evolution. Increased RAM could also help in storing large populations of deep symbolic expression trees.
>
> **“Are there plans to incorporate additional techniques, such as deep learning or domain-specific priors, to address these challenges … Could the framework benefit from integrating neural network components for further optimization”**
>
> This is a very interesting point. Future directions can definitely include joint optimization of symbolic trees using SymMatika and deep neural network-based symbolic regression methods as a hybrid system. Inspired by the commonly used optimization algorithm in deep neural networks such as gradient descent algorithms, another direction can leverage these algorithms for constant tuning for the SymMatika-found expression structures. However, these future directions are out of the scope of our current paper. We leave them for future work.
>
> **“How does SYMMATIKA's structural motif reuse compare to neural-guided symbolic regression methods in terms of scalability and interpretability”**
>
> SymMatika remains highly competitive in terms of scalability and interpretability. As shown in our experiments with comparisons with neural-guided methods, SymMatika surpasses most deep-learning based algorithms (e.g. SNIP, DSR, Fig. 3) in predictive accuracy (mean $R^2$>0.99 for Feynman and median $R^2$ for Black-box) and recovery rates across all tested benchmarks, and is capable of scaling to high-variable problems, e.g. 2-10 variables for Black-box; we have made this more explicit in the paper [lines 357]. SymMatika leads the Pareto front in complexity for the Feynman equations and surpasses algorithms with similar predictive accuracy by factors of 2-4.5 on Black-box, demonstrating SymMatika’s superior interpretability.
>
> **“While SYMMATIKA demonstrates impressive speed improvements over Eureqa, how does it compare in terms of computational resource requirements with other modern SR frameworks like PySR or TPSR”**
>
> SymMatika doesn’t require LLMs or GPUs [lines 366-367], and is computationally cheaper than neural-guided approaches, including TPSR. SymMatika has similar computational cost to PySR, but surpasses PySR on benchmark performance (Tab. 4, 5 in Appendix Sec. A.4).

---

> > ### Author Response · Authors · 2025-12-03
> > **Author Response Summary**
> >
> > We thank the reviewer again for the thoughtful comments, and appreciate the reviewer’s recognition of our dual support for explicit and implicit regression and the efficacy of our motif library through ablation studies and strong experimental results. In our response and revision of the paper, we clarified that our evaluations did in fact include both synthetic and real-world datasets within the Black-box benchmark suite and we highlighted this explicitly in our revised paper to avoid any ambiguity regarding real-world applicability. We addressed the concern regarding statistical significance by reporting rigorous testing for all baselines, explicitly noting cases where other methods did not provide sufficient data to support formal statistical analysis. We emphasized that SymMatika’s contribution extends beyond implicit-derivative metrics by introducing a unified SR algorithm supporting explicit and implicit regression, as well as a continuous feedback mechanism guiding evolutionary search. While our structural motif reuse is inspired in part by biological sequence analysis, the underlying mechanism for identifying high-impact subexpressions through fitness-contribution scoring is new in symbolic regression and advances GP-based search, as evidenced by our ablation studies and powerful experimental results. We discussed potential future extensions, such as enabling compositional motifs for deeply nested unary structures and exploring integrating neural components. Finally, we clarified SymMatika’s computational advantages and accessibility by not requiring LLMs or GPUs and still maintaining state-of-the-art performance.

---

### Meta-Review · Area_Chair_51TS · 2026-01-05

**Summary:**

SymMatika was proposed for both explicit and implicit SR using feedback-guided genetic programming with the reusable motif library. The authors claimed that SymMatika achieved state-of-the-art (SOTA) symbolic regression performances with better scalability on Nguyen, Feynman, SRBench, and Eureqa benchmarks, due to "continuous adaptation of the search distribution over symbolic expressions" by combining adaptive operator scheduling, motif-based recombination, and implicit-derivative fitness evaluation.

**Reviewer Concerns:**

Based on the available reviewer-author discussions, the authors have tried to emphasize the novelty of the SymMatika on considering implicit SR when addressing the novelty concern by reviewer C1gH but talked about "continuous adaptation of the search distribution" when responding to reviewer Ljt9. The authors may want to appropriately posit their corresponding contributions in their revision considering the existing research in genetic programming as stated by C1gH.

Some of the reviewers also requested theoretical foundations of the reported SymMatika performances but the authors did not provide clear answers to these questions by mostly referring to genetic programming performance guarantees. Reviewer FFxE asked for a more comprehensive Pareto-optimality analysis, which the authors might want to consider for their future submission.

Almost all reviewers have raised concerns on the inability of recovering certain complex Feynman equations and potentially biased experimental evaluation with selected expressions from the benchmarks. For fair comparison, the authors may also need to provide the corresponding motifs used in their library considering their claims on scalability and efficiency advantaged over benchmarked methods. Also, there are concerns considering robustness to the varying noise levels when evaluating on Nguyen and Feynman benchmarks. For example, Table 5 appears to be imported from LaSR (Symbolic Regression with a Learned Concept Library) Table 1, which was reported under a noised level (0.001). The authors may need to clearly describe their experimental settings as different noise levels may lead to very different recovery rate results.

Finally, both prediction and recovery performance comparison, the authors may want to evaluate other methods. For example, in RSRM: Reinforcement Symbolic Regression Machine (https://openreview.net/forum?id=PJVUWpPnZC), 100% recovery rate on Nguyen benchmark was reported.

**Reviewer Scores:**

Based on the available discussions, it is highly likely that the reviewers will keep their scores and reviewer C1gH and yMEA still have serious concerns regarding this submission.

---

### Decision · Program_Chairs · 2026-01-26

Reject